# Eltanexor Effectively Reduces Viability of Glioblastoma and Glioblastoma Stem-Like Cells at Nano-Molar Concentrations and Sensitizes to Radiotherapy and Temozolomide

**DOI:** 10.3390/biomedicines10092145

**Published:** 2022-08-31

**Authors:** Katharina Otte, Kai Zhao, Madita Braun, Andreas Neubauer, Hartmann Raifer, Frederik Helmprobst, Felipe Ovalle Barrera, Christopher Nimsky, Jörg W. Bartsch, Tillmann Rusch

**Affiliations:** 1Department of Neurosurgery, Philipps University Marburg, Baldingerstrasse, 35043 Marburg, Germany; 2Department of Hematology, Oncology & Immunology, Philipps University Marburg, Baldingerstrasse, 35043 Marburg, Germany; 3FACS Core Facility, Philipps University Marburg, Hans-Meerwein-Strasse 3, 35043 Marburg, Germany; 4Department of Neuropathology, Philipps University Marburg, Baldingerstrasse, 35043 Marburg, Germany

**Keywords:** GBM, Eltanexor, SINE, XPO1, temozolomide, radiosensitivity

## Abstract

Current standard adjuvant therapy of glioblastoma multiforme (GBM) using temozolomide (TMZ) frequently fails due to therapy resistance. Thus, novel therapeutic approaches are highly demanded. We tested the therapeutic efficacy of the second-generation XPO1 inhibitor Eltanexor using assays for cell viability and apoptosis in GBM cell lines and GBM stem-like cells. For most GBM-derived cells, IC_50_ concentrations for Eltanexor were below 100 nM. In correlation with reduced cell viability, apoptosis rates were significantly increased. GBM stem-like cells presented a combinatorial effect of Eltanexor with TMZ on cell viability. Furthermore, pretreatment of GBM cell lines with Eltanexor significantly enhanced radiosensitivity in vitro. To explore the mechanism of apoptosis induction by Eltanexor, *TP53*-dependent genes were analyzed at the mRNA and protein level. Eltanexor caused induction of *TP53*-related genes, *TP53i3*, *PUMA*, *CDKN1A*, and *PML* on both mRNA and protein level. Immunofluorescence of GBM cell lines treated with Eltanexor revealed a strong accumulation of CDKN1A, and, to a lesser extent, of p53 and Tp53i3 in cell nuclei as a plausible mechanism for Eltanexor-induced apoptosis. From these data, we conclude that monotherapy with Eltanexor effectively induces apoptosis in GBM cells and can be combined with current adjuvant therapies to provide a more effective therapy of GBM.

## 1. Introduction

Glioblastoma multiforme (GBM) is the most frequent malignant primary brain tumor in adults with a grim prognosis [1]. The latest data describe a five-year survival rate of less than 5% and median overall survival from diagnosis to death of around 11–15 months [2,3,4]. Furthermore, the health-related quality of life during disease progression is utterly poor [5,6]. Whilst other malignant entities benefited greatly from impressive scientific and clinical advancements within the past decades, the current GBM standard therapy is still quite conservative and foremost relies on surgery followed by adjuvant radio-chemotherapy with temozolomide (TMZ) [7,8]. Down the road, the established standard therapy as well as up-to-date clinical approaches with other cytostatics, targeted antibodies, immunotherapy, or tumor-treating fields ultimately fail due to the inevitable development of therapeutic resistance at various levels [9,10].

A significant number of mechanisms have been proposed in an attempt to explain the therapeutic resistance of GBM cells: e.g., the metabolic inactivation of drugs, increased drug efflux, increased DNA repair [11,12,13], loss of the potential of microglia to present antigens [14,15], induction of T-cell dysfunction [16,17,18], an immunosuppressive tumor microenvironment [19,20,21], the blood–brain barrier (BBB), and the role of corticosteroids regularly administered to GBM patients [22] effectively reducing the BBB permeability by reinforcement of endothelial tight junctions [23]. In addition, GBM stem-like cells (GSCs) have been proposed in recent years to be responsible for therapeutic resistance and tumor relapse [24].

The GBM tumor cell mass is composed of (i.) rapidly dividing cancer cells, forming the corpus of the tumor mass, and (ii.) self-renewing cells with stem cell characteristics are referred to as glioblastoma stem-like cells (GSCs) [25,26,27]. Whilst the former is sensitive to chemotherapy due to its significantly elevated proliferation rate, the latter is thought to be most resistant to the induction of apoptosis by the established standard therapy, bearing the capacity of tumor initiation as a source of glioma recurrence [8,13,28,29]. Therefore, novel efficient therapies targeting the resistance of GBM cells, especially GBM stem-like cells, to apoptosis are highly demanded. The main mechanism of cell death mediated by radiation therapy is not apoptosis, but rather a mixture of apoptosis, induction of senescence, autophagy, and necrosis [30]. Likewise, TMZ-generated DNA damage in GBM cells primarily induces cellular senescence [31,32,33]. Most GBM cells do even evade apoptosis upon the presence of TMZ, entering a senescent state that protects them against anticancer therapy. In addition, GBM cells may escape from senescence after TMZ therapy, contributing to the formation of recurrence [34]. 

Exportin 1 (XPO1) is a nuclear export protein responsible for the nuclear-cytoplasmic transport of hundreds of proteins and multiple RNA species, ensuring proper cellular functions [35]. Therefore, XPO1 is associated with an immense number of effects. Considering the diversity of RNA species exported by XPO1, it may have a profound impact on different aspects of RNA metabolism [36]. The vast number of proteins exported by XPO1 suggests functions within the regulation of mitosis [37,38], autophagy, biogenesis of the cytoskeleton, peroxisomes and centrosomes, ribosome maturation, translation, chromosomal structure, and mRNA degradation [39]; it is also required for microtubule nucleation [40], the assembly of mitotic spindles [41] and vesicle coats [36]. XPO1 is frequently overexpressed and deregulated in various human cancer entities (e.g., in sarcoma, DLBCL, multiple myeloma, KRAS-mutant lung cancer, pancreatic, ovarian, glioma, lung, gastric, prostate, and colorectal cancers) and is associated with poor prognosis [42,43,44]. Consecutively, the suppression of XPO1-mediated nuclear export has been postulated to present an apparent new therapeutic target strategy [36]. 

Novel selective inhibitors of nuclear export (SINE) demonstrated promising results in a wide range of cancer entities via extensive preclinical and clinical testing [45,46]. Several tumor suppressor proteins, including p53, BRCA1/2, and p27, are amongst the transported proteins of XPO1. The impaired export of tumor suppressors through SINEs is believed to be one of many potential mechanisms of action for XPO1 inhibitors [36,47,48]. Research showed that SINE compounds, such as the first-generation XPO1 inhibitor Selinexor (KPT-802), approved by the FDA for multiple myeloma therapy can induce apoptosis in GBM cells and respective animal models without overt evidence of neurotoxicity [49,50]. Eltanexor (KPT-8602) is a novel second-generation XPO1 inhibitor currently under investigation in preclinical and early clinical settings for several cancer entities (colorectal, multiple myeloma, myelodysplastic syndrome, ALL subtypes, etc.) [51,52,53,54]. Selinexor has been investigated for its potential in GBM cells as well as in early clinical trials. Data revealed improved radiosensitivity and efficacy as monotherapy, and suggest durable responses, disease stabilization, and progression-free survival [55,56]. So far, no data are available describing the effect of Eltanexor on GBM in vitro. 

Thus, the major goal of this work was to evaluate the therapeutic efficacy of Eltanexor in GBM cell lines and especially GBM stem-like cells. We sought to compare monotherapy as well as combining Eltanexor with the established standard therapy and to consecutively place our data into the context of the preexisting data for Selinexor.

## 2. Materials and Methods

### 2.1. Cell Culture

Experiments were performed with two established human glioblastoma cell lines (U87 and U251) and four primary GBM stem-like Cells (2017/74, 2017/151, 2016/175, and 2016/240). The U87 and U251 cell lines were obtained from ECACC, and their identity was verified by karyotyping and STR profiling. Cells were cultured in DMEM (11965092, Gibco™, Thermo Fisher Scientific, Waltham, MA, USA) containing 10% fetal bovine serum (FBS; S0615, Sigma-Aldrich, Munich, Germany), 1% Penicillin-Streptomycin (15140122, Gibco™, Thermo Fisher Scientific, Waltham, MA, USA), 1% Non-Essential Amino Acids (NEAA; 11140035, Gibco™, Thermo Fisher Scientific, Waltham, MA, USA), and 1% Sodium Pyruvate (11360070, Gibco™, Thermo Fisher Scientific, Waltham, MA, USA). The patient-derived GBM Stem-Like Cells (GSCs) were isolated as described previously [57] with approval from the ethics committee of the Faculty of Medicine, Philipps University Marburg (institutional review board number 185/11). GSCs were cultured in DMEM/F-12 + GlutaMAX™ (31331028, Gibco™, Thermo Fisher Scientific, Waltham, MA, USA) containing 2% B-27™ Supplement (12587010, Gibco™, Thermo Fisher Scientific, Waltham, MA, USA), 1% Amphotericin B (15290026, Gibco™, Thermo Fisher Scientific, Waltham, MA, USA), 0.5% HEPES (H0887, Sigma-Aldrich, Munich, Germany), and 0.1% Gentamycin (A2712, Biochrom GmbH, Berlin, Germany) with the addition of EGF (315-09, PeproTech, Hamburg, Germany), and bFGF (100-18B, PeproTech, Hamburg, Germany) in a final concentration of 0.02 ng/µL. All cultures were maintained at 37 °C in a humidified atmosphere containing 5% CO_2_.

Primary astrocytes were obtained from 1–2-day old postnatal *p53^−/−^* mice as described previously [58]. Astrocytes were cultured in 10 mL DMEM + GlutaMAX™ supplemented with 1% Penicillin-Streptomycin and 10% FBS at 37 °C. Astrocytes (2000 cells/well (5-day treatment), or 4000 cells/well (3-day treatment)) were seeded 24 h prior to treatment with 200 µL media with or without Eltanexor at varying doses in 96-well plates.

Neuronal progenitor cells (NPCs) were isolated according to the protocol of Deshpande et al. [59] using two adult *p53^−/−^* mice, 20 and 8 weeks old at the time of cell harvest. The neuronal progenitor cells were cultivated initially in flasks with neuronal stem medium (500 mL DMEM/F12, Thermo Fisher Scientific, Cat#12634010; 2% B27 supplement (50×), Thermo Fisher Scientific, Cat#17504044; 1.35% GlutaMAX™ (100×), Thermo Fisher Scientific, Cat#35050061; and 1% Penicillin Streptomycin, Thermo Fisher Scientific, Cat#15140122). To maintain their neuronal progenitor status, cells were treated every 4 to 5 days with EGF (20 ng/mL) (Immunotools, Cat#12343406) and FGF-β (5 to 8 ng/mL) (Immunotools, Cat#12343623). Adherent cultures were carried out according to the protocol [59], with some modifications. Additionally, 2 × 10^3^ or 4 × 10^3^ NPCs were seeded on poly-ornithine hydrochloride (15 µg/mL diluted in distilled deionized H_2_O, Thermo Fisher Scientific, Cat#P2533-100MG) and Laminin (5 µg/mL diluted in phosphate-buffered saline (PBS), Sigma-Aldrich, Cat#L2020-1MG) coated 96-well plates in neuronal stem medium supplemented with EGF/FGF-β.

### 2.2. Reagents

The chemotherapeutic agent Temozolomide (TMZ; S1237) and the XPO1 inhibitor Eltanexor (KPT-8602; S8397) were purchased from Selleck Chemicals, Houston, TX, USA, and diluted in DMSO (A3672, AppliChem GmbH, Darmstadt, Germany) to yield a 200 mM and 99.23 mM stock solution, respectively. 

### 2.3. Viability Assay

To evaluate cytotoxic effects, cells were seeded in 96-well plates 24 h before treatment with Eltanexor, TMZ, or DMSO as vehicle control. For seeding cells, the following cell densities were chosen: GBM cell lines U87 and U251; 2000 cells/well, and GSCs; 10,000 cells/well. Twenty-four hours after plating, cells were treated with the indicated concentrations. As a control, (100% viability), the solvent DMSO was used in the concentration corresponding to the highest dose of Eltanexor. For cell lines U87 and U251, cell viability was detected after five days and for GSCs ten days after treatment. For this purpose, 50 µL CellTiter-Glo^®^ 3D Cell Viability Assay (G9682, Promega GmbH, Walldorf, Germany) was added to the well. After 15 min of mixing on a platform shaker and 15 min of incubation at room temperature, both in the dark, luminescence was measured using a FLUOstar OPTIMA Microplate Reader (BMG LABTECH, Ortenberg, Germany). Data were normalized to the control group.

### 2.4. Apoptosis Assay

Apoptosis was assessed using the Caspase-Glo^®^ 3/7 Assay (G8090, Promega GmbH, Walldorf, Germany) and Caspase-Glo^®^ 3/7 3D Assay (G8981, Promega GmbH, Walldorf, Germany), utilizing a luminogenic caspase-3/7 substrate which contains the tetrapeptide sequence DEVD in a reagent optimized for caspase activity, luciferase activity, and cell lysis. Cells were seeded in 96-well plates at a density of 4000 cells/well for U87 and U251 and at a density of 10,000 cells/well for GSCs. 24 h after seeding, cells were treated with the drugs. For U87 and U251, apoptosis was measured 24 h after treatment. For GSCs, apoptosis was measured 48 h after treatment. For this, 20 µL of the reagent was added to each well (Caspase-Glo^®^ 3/7 Assay for U87 and U251, and Caspase-Glo^®^ 3/7 3D Assay for GBM Stem-Like Cells). The contents were mixed gently on a platform shaker for 30 s, followed by a one-hour incubation at room temperature in the dark. Luminescence was recorded using a FLUOstar OPTIMA Microplate Reader (BMG LABTECH, Ortenberg, Germany). Equivalent to the viability assay, data were normalized to the control group.

### 2.5. FACS Analysis 

Apoptosis FACS staining was determined using the eBioscienceTM Annexin V Apoptosis Detection Kit APC (88-8007-72, Invitrogen, Waltham, MA, USA). Briefly, U87 and U251 GBM cell lines were seeded at a density of 1 × 106 cells in T25 flasks overnight, refreshed with a medium containing 100 nM Eltanexor or the same volume of DMSO as vehicle control for 24 h. All steps were then executed according to the manufacturer’s instructions. Cells were washed with ice-cold PBS and 1× binding buffer, followed by the addition of 5 μL Annexin V APC to 100 μL of cell suspension and incubation for 15 min at RT in the dark. Cells were washed in 1× binding buffer and resuspended in 200 μL binding buffer. After adding 5 μL propidium Iodide, incubation was performed for 30 min at RT in the dark. Finally, cells were washed in 1× binding buffer and then resuspended in 400 μL binding buffer for FACS measurement. 

Cell cycle analysis was performed using Propidium Iodide (PI) (P4170, Sigma, Dreieich, Germany). U87 and U251 GBM cell lines were seeded at a density of 1 × 10^6^ cells in T25 flasks overnight, refreshed with a medium containing either 100 nM Eltanexor or the same volume of DMSO control for 24 h. Cells were collected, washed with ice-cold PBS, and fixed with 80% cold ethanol overnight. Cells were then washed with PBS and resuspended in 1 mL PI buffer containing 0.1% Triton X-100 (T8787, Sigma, Dreieich, Germany) and 1 mg DNase-free RNase A. Incubation was performed for 30 min at RT in the dark. Finally, cells were washed in 1 mL PBS and resuspended in 400 μL PBS for measurements.

### 2.6. Radiation Exposure

Cells were irradiated with X-rays using a Precision X-RAD 320ix biological irradiator (Precision X-Ray, North Branford, CT, USA) at 320 kV and 8 mA, dose rate of 1.0 Gy/min, filter: 0.5 mm Al/0.5 mm Cu. Absolute dose measurements confirmed the applied doses.

### 2.7. Colony Formation Assay

For U87 and U251 clonogenic survival was determined by performing the colony formation assay. Cells were plated in 6-well plates with appropriate cell numbers (ranging from 200 to 6000 cells/well depending on cell line and radiation dose). 24 h after seeding, Eltanexor was applied in doses of 50 nM (U87) and 100 nM (U251), respectively. After one hour of incubation, cells were irradiated with 2, 4, and 6 Gray. The following day, 24 h after irradiation, the media was replaced with fresh drug-free media. U251 cells were left to grow for 10 days to form colonies and U87 cells for 14 days. Cells were then fixed with 10% formalin (F8775, Sigma-Aldrich, Munich, Germany) for 15 min and stained with 0.1% crystal violet (C.I. 42555, Merck KGaA, Darmstadt, Germany) for 30 min. Colonies consisting of at least 50 cells were counted and surviving fractions were calculated. Survival curves were created after normalizing for the cytotoxicity induced by Eltanexor. Dose enhancement factors (DEF) at different radiation doses were calculated by dividing the mean survival of the control group by the mean survival of the treatment group.

### 2.8. Immunofluorescence Staining

Cells were seeded on Collagen Type I (1:20 dilution; C7661, Sigma-Aldrich, Munich, Germany) coated coverslips at a density of 100,000 cells/well in a 24-well plate. GSC cells were seeded on laminin (5 µg/mL diluted in PBS) coated coverslips. After overnight incubation, cells were washed 3 times with PBS (D8537, Sigma-Aldrich, Munich, Germany), fixed with 4% paraformaldehyde (PFA) for 15 min, and permeabilized with 0.3% Triton X100 (T8787, Sigma-Aldrich, Munich, Germany) for 15 min, followed by 1 h blocking with 5% Bovine Serum Albumin (BSA; A7030, Sigma-Aldrich, Munich, Germany) to avoid non-specific binding. Cells were incubated with the primary antibodies anti-p21 (1:250 dilution in BSA; 10355-1-AP, Proteintech, Manchester, UK), anti-p53 (1:2000 dilution in BSA; 2524, Cell Signaling Technology, Leiden, The Netherlands), and anti-PIG3 (TP53I3; 1:150 in BSA; CF503656, OriGene Technologies GmbH, Herford, Germany) overnight at 4 °C. The next day, cells were washed 3 times with PBS and incubated with the secondary antibodies Donkey Anti-Rabbit DyLight 488 (dilution 1:250 in 5% BSA; ab96919, Abcam, Cambridge, UK) and Donkey Anti-Mouse DyLight 488 (dilution 1:250 in 5% BSA; ab98794, Abcam, Cambridge, UK) for 1 h at RT in the dark. Nuclei were stained with Hoechst 33342 (1:10,000 dilution; Cat. No. 62249, Thermo Fisher Scientific, Waltham, MA, USA) for 15 min at RT in the dark before covering with anti-fade mounting medium (S3023, Agilent, Santa Clara, CA, USA). Images were obtained using a Keyence BZ-X800 microscope (Keyence Deutschland GmbH, Neu-Isenburg, Germany).

### 2.9. RNA Isolation, Reverse Transcription, and Quantitative Real-Time PCR

RNA Isolation was performed as described previously [60]. Briefly, total RNA was isolated by QIAzol Lysis Reagent (Cat. No. 79306, Qiagen GmbH, Hilden, Germany) and absorbance was measured with OD 260/280 ratio between 1.8 and 2.1. About 2 µg of RNA was reverse transcribed into cDNA with RNA to cDNA EcoDry™ Premix kit (Takara Bio Inc., Kusatsu, Japan) according to the manufacturer’s instructions. PCR amplification reactions were carried out in 20 µL total reaction volumes with 2 µL cDNA, 2 µL primers synthesized by Qiagen GmbH (Hilden, Germany), 6 µL nuclease-free water, and 10 µL SYBR Green/Rox Master Mix (Primer Design, Southhampton, UK) in a StepOnePlusTM Real-Time PCR system (Thermo Fisher Scientific, Waltham, MA, USA). The qPCR protocol set initial denaturation at 95 °C for 10 min, followed by 40 amplification cycles at 95 °C for 15 s and 60 °C for 1 min. For the housekeeping control gene, we used RPLP0 XS13fw: 5′-TGGGCAAGAACACCATGATG-3′; XS13rev: 5′- AGTTTCTCCAGAGCTGGGTTGT-3′; for *TP53*: p53fw: 5′-ACCACCATCCACTACAACTACAT-3′; p53rev: 5′-CCAGGACAGGCACAAACA-3′ (Microsynth SeqLab GmbH, Goettingen, Germany). All other primers for *TP53*-dependent genes were purchased from Qiagen (Quantitect, Hilden, Germany): BAX Product Name: Bax (Hs_BAX_1), GeneGlobe ID: QT00031192; TP53AIP1 (Hs_TP53AIP1),_GeneGlobe Id: QT02377634; PIG3 (Hs_TP53I3), GeneGlobe Id: QT00010332; PIDD (Hs_PIDD1),_GeneGlobe Id: QT02406691; PUMA (Hs_BBC3),_GeneGlobe Id: QT00082859; NOXA (Hs_PMAIP1),_GeneGlobe Id: QT01006138; CDKN1a (Hs_CDKN1A), GeneGlobe Id: QT00062090; PML (Hs_PML), GeneGlobe Id: QT00090447. The fold changes in gene expression relative to control were calculated using the 2^−ΔΔCT^–method.

### 2.10. Protein Extraction and Western Blot Analysis

Cells were washed 3 times with ice-cold PBS. Total protein extraction was performed with RIPA buffer (50 mM HEPES pH 7.4; 150 mM NaCl; 1% (*v*/*v*) NP-40; 0.5% (*w*/*v*) Natriumdeoxycholate; 0.1% (*w*/*v*) SDS; 10 mM Phenantrolin; 10 mM EDTA; PierceTM Protease Inhibitor Mini Tablets, EDTA-free, Thermo Fisher Scientific, Waltham, MA, USA; PierceTM Phosphatase Inhibitor Mini Tablets, Thermo Fisher Scientific, Waltham, MA, USA). Subsequently, protein lysates were boiled for 5 min in Laemmli (60 mM Tris-HCl pH 6.8; 2% (*w*/*v*) SDS; 10% (*w*/*v*) Glycerol; 5% (*v*/*v*) ß-Mercaptoethanol; 0.01% (*w*/*v*) Bromophenol-Blue) and sample reducing buffer (B0009, InvitrogenTM, Thermo Fisher Scientific, Waltham, MA, USA). After protein separation by SDS Page using a 12.5% polyacrylamide gel, proteins were transferred to a nitrocellulose membrane (A29591442, GE Healthcare Life science, Solingen, Germany) and blocked with 5% non-fat milk (T145.3, Carl Roth GmbH + Co. KG, Karlsruhe, Germany) for 1 h at RT. For the detection of proteins, membranes were incubated overnight at 4 °C with the following primary antibodies: cleaved caspase-3 (1:1000 dilution in 5% milk, 9661, Cell Signaling Technology, Leiden, The Netherlands), p53 (1:1000 dilution in 5% milk in TBST, 2524, Cell Signaling Technology), CDKN1A/p21 (1:2000 dilution in 5% milk in TBST, 10355-1-AP, Proteintech, Manchester, UK), TP53i3/PIG3 (1:2000 dilution in 5% milk in TBST, CF503656, OriGene Technologies GmbH, Herford, Germany), PUMA (Abcam ab 9643, Cambridge, UK; 1:2000 in 5% milk in TBST), PML (1:1000 in 5% milk in TBST, Proteintech Cat. No. 21041-AP), and β-Tubulin (1:2000 dilution in 5% milk in TBST, NB600-936, Novus Biologicals, Littleton, CO, USA). The next day, nitrocellulose membranes were incubated with the secondary antibodies Donkey Anti-Mouse (HRP) (dilution: 1:4000 in 5% milk in TBST; ab97030, Abcam, Cambridge, UK) and Donkey Anti-Rabbit (HRP) (dilution: 1:4000 in 5% milk in TBST; ab97064, Abcam, Cambridge, UK) for 1 h at RT. After washing the membranes with TBST, the detection was performed by utilizing the ChemiDoc MP Imaging System (Bio-Rad Laboratories GmbH, Feldkirchen, Germany).

### 2.11. Migration Assay 

Migration experiments were performed by scratch assays. For this, 10,000 cells were seeded in 24-well plates overnight. For starvation, DMEM supplemented with 0.5% (*v*/*v*) FBS was added 12 h prior to the scratch. In each well, a gap was scratched at the bottom of the well using a 20 uL pipette tip, and cells were washed with a normal growth medium to remove non-adherent cells. Afterwards, fresh DMEM with 10% FBS was added either containing Eltanexor or DMSO as control. Images at each edge of the gap were taken at time points 0 h, 6 h, and 24 h. All images were analyzed using Image J software to determine cell numbers in the gap, respectively.

### 2.12. Data Analysis

Data from multiple replicates were presented as mean ± SD. Statistical analyses were performed using GraphPad Prism version 9.1.0 for macOS (GraphPad Software, San Diego, CA, USA) and the results were considered as not significant (ns, *p* > 0.05). Significance values are shown by asterisks with * (*p* < 0.05), ** (*p* < 0.01), *** (*p* < 0.001), and **** (*p* < 0.0001). One-way analysis of variance (ANOVA) was used for the statistical analysis of multiple comparisons. Unpaired Student’s *t*-test was applied for statistical comparison among two groups. The half-maximal inhibitory concentrations (IC_50_) were determined by a non-linear regression method using least square fit (GraphPadPrism).

## 3. Results

### 3.1. Reduced Cell Viability of GBM Cells after Treatment with the XPO1 Inhibitor Eltanexor

As shown for various tumor entities, inhibition of the karyopherin exportin-1 (XPO1), a nuclear transport receptor interacting with the leucine-rich nuclear export signal (NES), can have a profound effect on the survival of tumor cells. This is particularly relevant as XPO1 expression is significantly higher in GBM tissue compared to normal brain tissue (Appendix A). Whereas the first-generation XPO1 inhibitor, Selinexor, was tested in GBM cells, no data so far are available for the more efficient second-generation XPO1 inhibitor, Eltanexor, which should be tested in GBM cell lines and in glioblastoma stem-like cells (GSCs), which are the most resistant to therapeutic interventions and are considered as the origin of recurrent glioblastoma after acquiring therapy resistance. It is noteworthy that the cells we investigated induce XPO1 mRNA expression after the addition of Eltanexor (Appendix A), indicating that treated cells activate a compensatory mechanism to overcome the Eltanexor effects, but fail to do so. 

To test the therapeutic efficacy of Eltanexor in terms of cell death induction, the GBM cell lines U87 and U251 and four patient-derived GSCs were treated with Eltanexor spanning a concentration range from 1 nM to 10 μM. Cell viability of treatment groups or vehicle (DMSO) control was determined after five days for GBM cell lines and after 10 days for GSCs and was normalized to DMSO controls (Figure 1A–F).

For all cell lines and the GSCs, IC_50_ values of Eltanexor are below 300 nM, in particular for GSCs, IC_50_ values are even below 200 nM. These data demonstrate that Eltanexor is very effective in reducing the viability of GBM cells. To evaluate the efficacy of Eltanexor in non-malignant brain cells, viability was assessed in primary astrocytes and neuronal precursor cells (Appendix A). For primary astrocytes, the IC_50_ value for Eltanexor is ten times higher than for GSC74 (383 nM). For neuronal progenitor cells, IC_50_ values are similar to those for GSCs (57.8 nM). We found a complete loss of cell viability after treatment with Eltanexor and hypothesized that the cells are eliminated by apoptosis, similar to the cell death mechanism described for the first-generation XPO1 inhibitor Selinexor. Apoptosis was evaluated by caspase-3 analysis using a luminogenic caspase-3 substrate (DEVD peptide), caspase-3 western blot, and annexin V staining in FACS analyses using GBM cell lines and the most sensitive GSC cell line, GSC_74 (Figure 2). All results support the notion that Eltanexor induces apoptosis in GBM cells. 

### 3.2. Eltanexor Induces Apoptosis in GBM Cells by Increased TP53 Signaling

To analyze the mode of cell death in GBM cell lines and the two most responsive GSC lines GSC 74 and GSC 240, activity levels of the pro-apoptotic protease caspase-3 were determined 24 h after the addition of varying doses of Eltanexor for GBM cell lines and 48 h a for GSCs (Figure 2). After 24 h (U87,U251) or 48 h (GSC 74) of treatment with 10, 100, and 500 nM Eltanexor, significantly increased caspase-3 activities were observed. In most cases, cells responded with caspase-3 activation to an Eltanexor dose of 100 nM, in particular glioma stem-like cells (Figure 2C,F). In accordance with caspase-3 activity, western blot data revealed increased concentrations of cleaved caspase-3, reflecting the extent of caspase-3 activation (Figure 2D–F). Furthermore, increased Annexin V staining was observed after Eltanexor treatment in U87 (Figure 2G,H) and U251 cells (Figure 2I,J). Induction of apoptosis is accompanied by the arrest of the S phase, i.e., fewer cells in the S phase after Eltanexor treatment, as revealed by cell cycle analysis using FACS sorting (Appendix A). 

As Eltanexor induces changes in the cell cycle and apoptosis of GBM cells, we next investigated the downstream mechanism of XPO1 inhibition by analyzing *TP53* and *TP53*-related gene regulation affected by Eltanexor in GBM cells. Accordingly, GBM cells were treated with 100 nM Eltanexor for 24 h, GSC 74 cells were treated with the identical concentration for 48 h, and the mRNA expression levels of *TP53*, *BAX1*, *Tp53AIP1*, *Tp53i3*, *BBC3*, *PMAIP1*, *CDKN1A (p21)*, *PIDD1*, and *PML* were determined by qPCR (Figure 3).

The most consistent mRNA expression changes were observed for *TP53*, *Tp53i3*, *CDKN1A*, and *PML* (Figure 3A). Increased mRNA expression levels of PUMA (p53 upregulated regulator of apoptosis) were only observed in GSC cells. The corresponding protein analyses were performed for p53, TP53i3, PUMA, CDKN1A, and PML in Eltanexor treated GBM cell lines U87 and U251 and in the GSC 74 line by western blotting (Figure 3B,C). A significant increase in p53 protein was detected in U87, U251, and GSC 74 cells (Figure 3 and Appendix A). The relatively high protein levels of p53 were not affected by the serum, as we compared p53 levels in U87 and U251 cells after serum starvation and found no significant differences (Appendix A). In addition, Eltanexor caused induction of CDKN1A in U87 cells, whereas GSC cells showed the highest induction of PUMA in agreement with the mRNA analysis. No induced protein levels of Tp53i3 were seen in the cells investigated. PML was upregulated by Eltanexor in all cells investigated (Figure 3C). Besides the described effects of Eltanexor, lower concentrations of Eltanexor are able to inhibit cell migration as a result of a series of scratch assays performed with U87 and U251 cells. After 6 h for U87 and 24 h for U251, cell migration was significantly reduced (Appendix A). 

### 3.3. Eltanexor Causes p53 and CDKN1A to Be Retained in the Nucleus of GBM Cells 

We further tested if Eltanexor can cause the accumulation of p53-related proteins in the cell nucleus. We tested CDKN1A, TP53i3, and p53 in U87 and U251 cells, and CDKN1A and p53 in GSC cells after treatment with Eltanexor. Immunofluorescence was performed with antibodies detecting CDKN1A, Tp53i3, and p53 in control (vehicle) and Eltanexor treated cells to assess possible changes in the cellular distribution of the proteins. Cell nuclei were counterstained with DAPI (Figure 4) and the fluorescence signals detected in nuclei vs. cytoplasma were quantified and calculated as the ratio, so that all values >1 represent protein accumulation in the cell nucleus. In GBM cells investigated, we found a nuclear accumulation of CDKN1A and p53 (Figure 4D) after Eltanexor treatment. The U87 and U251 cells revealed additional nuclear staining of TP53i3 which was not seen in GSC 74 cells. 

### 3.4. Co-Treatment of Eltanexor with Temozolomide (TMZ)

We next investigated if a combination treatment of GBM cells with TMZ and Eltanexor increases the efficacy of TMZ. In both GBM cell lines, co-treatment of TMZ with Eltanexor had a significant effect on the cell viability (Figure 5A–D). 

In particular, in U251 cells less sensitive to Eltanexor co-treatment with TMZ caused a significant reduction in cell viability. In all other cell lines investigated, only the low dose of Eltanexor showed an effect when combined with TMZ, however, this effect was overruled by higher concentrations of Eltanexor (100 nM) in all Eltanexor-responsive cell lines (U87, GSCs 74 and 240), but not in U251 cells. 

The fact that cell viability cannot be reduced when comparing 100 nM Eltanexor with the combined TMZ/100 nM Eltanexor treatment (Figure 5) suggests that both drugs might induce similar mechanisms of cell death so that there is no further effect on cell viability when combining these two drugs. Alternatively, it can be argued that cell death induced by Eltanexor is much faster than the cell death induced by TMZ, as these are kinetically different with 24 h for Eltanexor vs. five days for TMZ when cells undergo senescence-like processes that ultimately lead to cell death.

### 3.5. Eltanexor Sensitizes GBM Cells to Radiotherapy 

A different mode of cell death involving DNA double-strand break repair and p53 activation can be observed when cells are subjected to irradiation with photons, equivalent to radiotherapy of patients. Therefore, we combined Eltanexor treatment with irradiation at increasing doses and analyzed cell viabilities in U87 and U251 GBM cells (Figure 6).

For U87 and U251 cells, Eltanexor sensitizes cells to X-ray radiotherapy with low significance in U87 (average DEF overall doses = 1.26) and high significance in U251 cells (average DEF overall doses = 1.73). 

## 4. Discussion

In our study, we demonstrated for the first time to our knowledge the efficacy of Eltanexor in GBM cell lines and, in particular, in difficult-to-treat GBM stem-like cells. We describe nanomolar IC_50_ values for GBM cell lines and even lower values for patient-derived GBM stem-like cells. Compared to TMZ, IC_50_ values for Eltanexor in GBM cell lines are lower by several orders of magnitude. Thus, especially for GBM stem-like cells, Eltanexor proves to be effective on GBM cell lines and as such, could be more potent with an average IC_50_ value of 82 nM compared to Selinexor with an average IC_50_ of 133 nM [49]. In contrast to the work of Green et al., we analyzed GBM cell lines and GBM stem-like cells rather than primary GBM cell lines. There are no comparable data available for monotherapy with Selinexor on GBM stem-like cells. 

Considering that U87 cells exert a higher IC_50_ value of ~250 μM for TMZ, it is remarkable that IC_50_ values for Eltanexor monotherapy are less than 100 nM. In contrast, U251 cells, known to be more sensitive to TMZ (IC_50_: ~50 μM) show the highest IC_50_ of all GBM cells tested for Eltanexor. This difference might be linked to the *p53* status of U87 with wild-type *p53* vs. U251 cells mutated in *p53*, assuming that the major mechanism induced by Eltanexor is apoptosis. These data are in agreement with a previous study using Selinexor in GBM therapy [49] and given XPO1 as a target, we do not expect an alternative mechanism of cell death induced by Eltanexor. We hypothesized that the reported nuclear accumulation of p53 could cause transcriptional changes in p53-dependent genes, and a gene set of eight genes was analyzed by qPCR. Increased transcription levels were observed for *Tp53*, *Tp53i3*, *PUMA*, *CDKN1A* (*p21*), and *PML.* We confirmed the increased abundance of these proteins in GBM cell lines and a representative GBM stem-like cell line. 

Mechanistically, we hypothesize that p21 and p53 are critical elements targeted by XPO1 inhibition using Eltanexor, as these proteins accumulate in the cell nucleus. Although it is clear that p21 is a transcriptional target of p53, its function is ambiguous. It was reported that the function of p21 depends on the cell type. In most cancer cell lines, p53-dependent p21 induction is essential for cell cycle arrest, but in some, p21 is dispensable [61]. Further studies have reported that nuclear accumulation of p53 and the consequential induction of CDKN1A/p21 by p53 causes the induction of apoptosis in embryonic stem cells [62]. As we detected a strong nuclear accumulation of CDKN1A/p21 following Eltanexor treatment in GBM cell lines, we can assume that a similar mechanism might apply. As another downstream target of p53, Tp53i3/PIG3 was induced by Eltanexor treatment and found accumulated in the nucleus of GBM cell lines, although to a lesser extent than CDKN1A/p21. TP53i3 is associated with DNA repair and reactive oxygen species-induced apoptosis [63] suggesting that Eltanexor induces apoptosis by p53-induced pro-apoptotic proteins in GBM cells. This mode of cell death occurs within 48 h and is distinct from the mode we observe for temozolomide. Notably, when combinations of Eltanexor (10 and 100 nM) with the respective IC_50_ concentrations for TMZ were used, only minor effects of TMZ were observed. This observation could be due to the fact that TMZ induces a cell cycle arrest and a slow transition of treated cells into senescence, rather than apoptosis. Since the effect of XPO1 inhibition is fast, apoptosis would be expected to be the speed-limiting step when both compounds are combined. In contrast, the combination of Eltanexor with x-ray radiation has a clear effect and suggests that Eltanexor acts on the DNA damage-induced p53 pathway induced by radiation as observed for Selinexor [64].

Our data in conjunction with data from other studies demonstrated better tolerability of non-malignant cells (astrocytes) and higher efficacy of Eltanexor compared to Selinexor in different malignancies including leukemia [63]. 

Several ongoing clinical trials are evaluating the efficacy, safety, and tolerability of Selinexor and Eltanexor in a variety of cancer entities (e.g.: multiple myeloma, myelodysplastic syndrome, colorectal cancer, acute myeloid leukemia, and others) [52,53,65,66]. Improved progression-free survival, overall survival, clinical benefit, stabilization, and control of disease support the novel approach of inhibiting XPO1 for the treatment of these and potentially other tumor entities. This is especially encouraging for application in a disease as complicated to treat as GBM. 

It has been reported that Selinexor is supposed to be more permeable to the BBB than Eltanexor [65]. Concerning the systemic treatment of other diseases, not located within the BBB, that led to the hypothesis that Eltanexor would be more applicable assuming a lower profile of brain-mediated side effects like nausea, decreased appetite, hyponatremia, and fatigue [52,66]. At first glance, Eltanexor seems to be the wrong choice for systemic treatment of patients with GBM concerning drug delivery to the tumor location. Based on our work and other studies, Eltanexor seems to be superior to Selinexor, as it shows significantly improved therapeutic efficacy in the clinical treatment of patients with refractory multiple myeloma, acute myeloid leukemia [52,67,68], and in preclinical GBM cell models. In this regard, we were able to establish the assumption of improved efficacy of Eltanexor in GBM cell lines and especially GBM stem-like cells compared to preexisting data for Selinexor [49]. Therefore, considering a way to bypass the BBB and administer Eltanexor directly to the tumor tissue, it can be speculated that Eltanexor may be more beneficial for GBM patients. The issue of drug delivery, especially direct application into the brain, is being addressed by current research and provides promising results [69]. 

Eltanexor showed reliable induction of apoptosis in GBM and GBM stem-like cells and the enhancement of radiosensitivity and possible combinatorial effects with TMZ allow the prospect of further clinical investigation in combination with the established standard treatment. Especially concerning radiotherapy, pretreatment with Eltanexor could be an opportunity to either reduce the radiation dose and side effects caused by radiation or enhance the efficacy of radiation. Further clinical evaluation and trials addressing the combination with established therapies may be reasoned based on our data.

## 5. Conclusions

Eltanexor as a second-generation XPO1 nuclear export inhibitor is highly efficient in monotherapy and the enhancement of radiosensitivity of GBM cell lines and GBM stem-like cells. Possible combinatorial effects with TMZ are cell-type dependent. Eltanexor acts most likely through a p53-dependent induction of genes involved in a fast cellular apoptosis process. Even though Eltanexor is believed to be less brain-barrier permeable than Selinexor, the existing data propose preclinical and clinical superiority concerning therapeutic efficacy, similar to other cancer types, in GBM. Further clinical investigations in combination with standard therapy and with regard to delivery routes are highly justified based on our work.

## Figures and Tables

**Figure 1 biomedicines-10-02145-f001:**
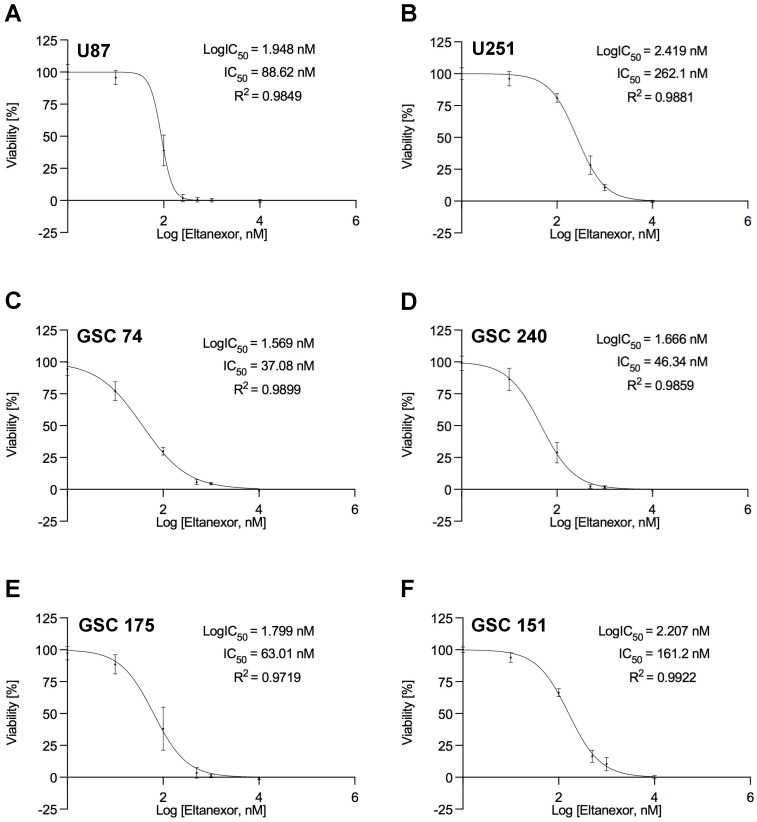
Dose-response curves for Eltanexor for U87 (**A**), U251 (**B**), and four patient-derived GSCs 74 (**C**), 240 (**D**), 175 (**E**), and 151 (**F**). All values are based on triplicate measurements performed in three independent experiments. Cell viability was determined using CellTiter-Glo reagent. IC_50_ values were calculated using non-linear regression with the least-square fit. Values (in nM) and the respective regression coefficients are noted in the respective diagrams.

**Figure 2 biomedicines-10-02145-f002:**
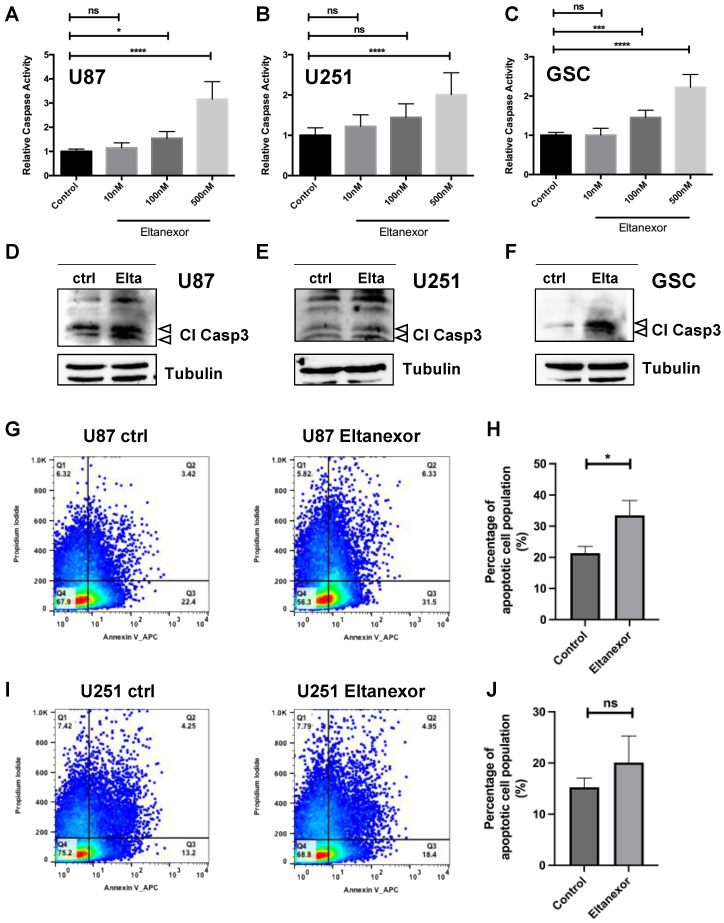
Detection of apoptosis in GBM cell after treatment with Eltanexor. A-C, caspase activities determined by Caspase GLO™ assays after treatment with 10, 100, and 500 nM Eltanexor for 24 h in U87 (**A**), U251 (**B**), and for 48 h in patient-derived GSCs 74 (**C**) compared to solvent DMSO (“ctrl”). All values are based on triplicate measurements performed in three independent experiments. Activities are shown relative to the control condition (set to 1) with relative changes given. (**D**–**F**), Western Blot to detect cleaved caspase-3 after treatment with 500 nM Eltanexor. Bands representing cleaved caspase-3 are marked by arrowheads (19 and 17 kD). (**G**–**J**), Representative Annexin V staining in U87 (**G**,**H**), and in U251 (**I**,**J**) cells as determined by FACS analyses. (**H**,**J**), Quantitative evaluation of annexin V staining from 3 consecutive experiments after treatment of either U87 (**H**) or U251 (**J**) cells with Eltanexor. Values are shown as mean values ± S.D. Significance was determined by one-way ANOVA with *, *p* < 0.05, ***, *p* < 0.001, and ****, *p* < 0.0001.

**Figure 3 biomedicines-10-02145-f003:**
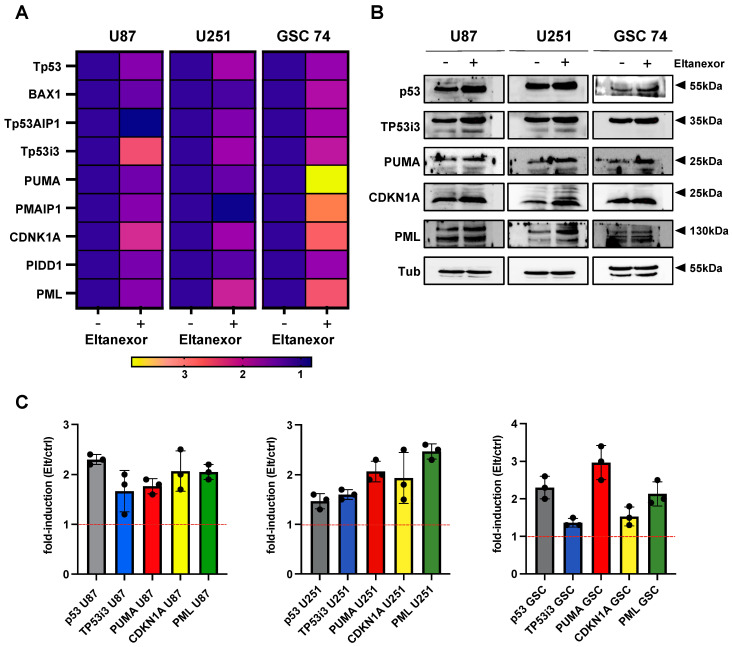
(**A**) Changes in gene expression induced by 100 nM Eltanexor (indicated by “+”) or the corresponding volume of vehicle (DMSO) treated for 24 h in U87, U251, and GSC 74 cells. All values are shown as color codes and are based on triplicate measurements performed in three independent experiments. All expression levels are shown relative to the control condition (“Eltanexor”, set to 1) with relative changes given. (**B**) representative images from western blot for induction of proteins p53, Tp53i3, PUMA, CDKN1A, and PML after treatment of U87, U251, and GCS74 cells with Eltanexor (100 nM) for 24 (U87, U251) and 48 (GSC74) hours. (**C**) quantification of immunoblots based on three independent replicates relative to the control condition (set to 1, red dotted line).

**Figure 4 biomedicines-10-02145-f004:**
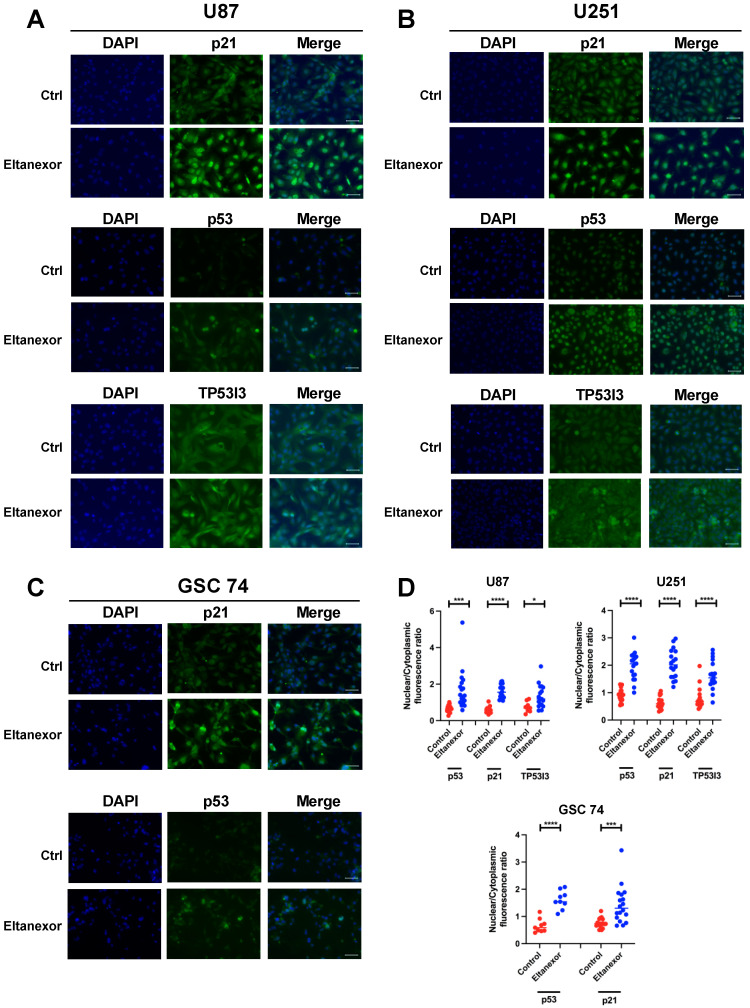
Immunofluorescence of U87 (**A**) and U251 (**B**) cells for CDKN1A, p53, and Tp53i3 and of GSC 74 cells (**C**) for CDKN1A and p53 after treatment with Eltanexor (100 nM) or the corresponding volume of vehicle (DMSO, “Ctrl”) for 24 h. Staining with either DAPI, antibodies, or merged images. Overlay of nuclear signals with DAPI is visible in Eltanexor treated cells. Scale bar, 50 μm, valid for all images. (**D**), quantification of the nuclear to cytoplasmic ratio for fluorescence signals detected as revealed by counting at least 9 individual cells (*n* dots). All ratios >1 denote nuclear accumulation. Significances were determined by Student’s *t*-test with *, *p* < 0.05, ***, *p* < 0.001, and ****, *p* < 0.0001.

**Figure 5 biomedicines-10-02145-f005:**
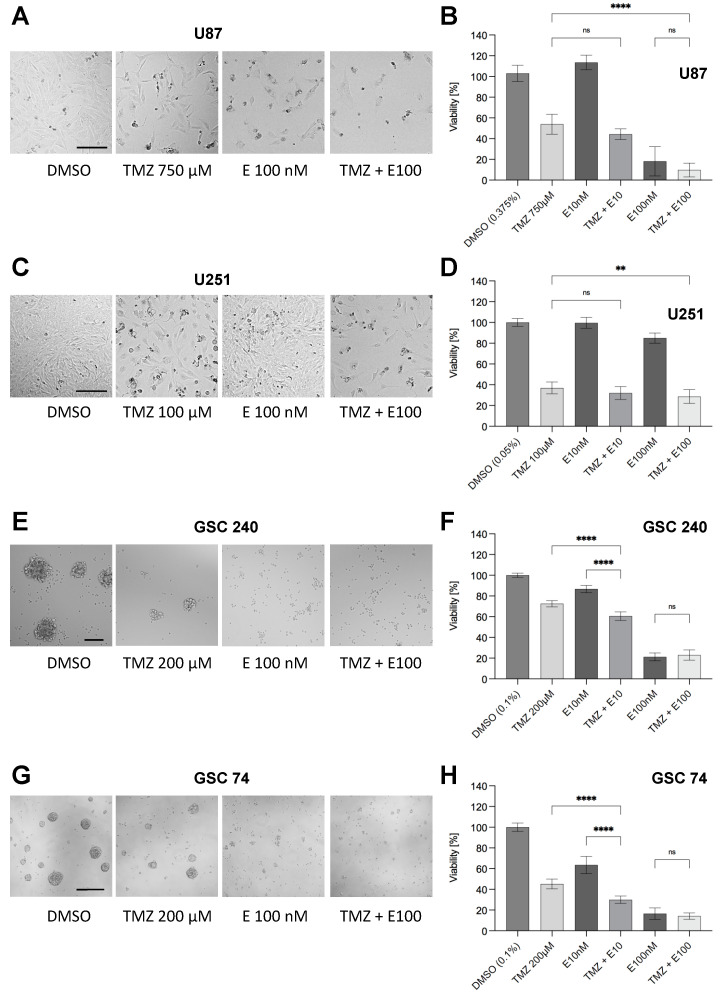
Effect of co-treatment TMZ/Eltanexor (10 and 100 nM) and temozolomide IC_50_ concentrations determined for each cell line (750 μM for U87, 100 μM for U251, and 200 μM for GSC cells, respectively) on viability of GBM cell lines U87 and U251 and GSC cells (2016/240 and 2017/74) after 5 days for cell lines and 10 days for GSC cells. Left panel (**A**,**C**,**E**,**G**), representative bright-field micrographs demonstrating cell loss after treatment. Right panel (**B**,**D**,**F**,**H**), quantification of cell viability using CellTiter-Glo^®^. All viabilities are shown relative to DMSO as solvent corresponding to the TMZ + 100 nM Eltanexor condition. Values are provided as a result of three independent experiments performed in triplicates. Statistical significance was determined using one-way ANOVA with **, *p* < 0.01, and ****, *p* < 0.0001. Scale bars in (**A**), valid for all images, 200 μm.

**Figure 6 biomedicines-10-02145-f006:**
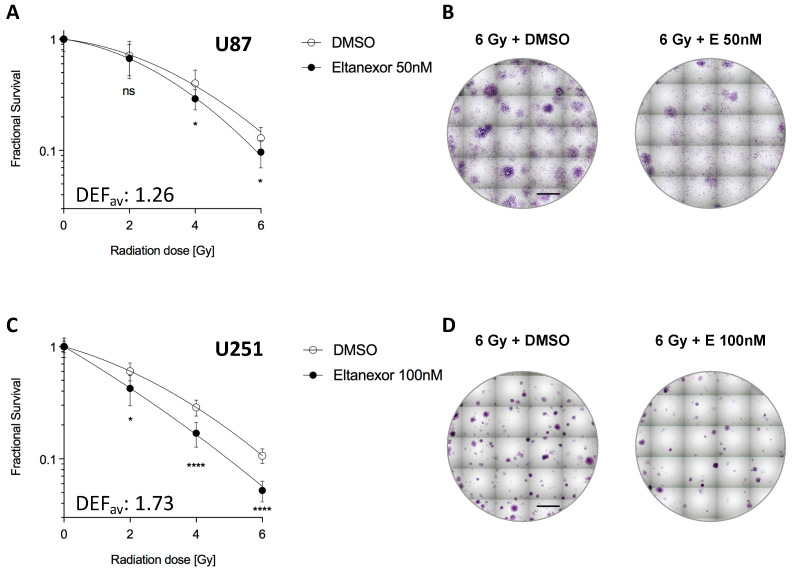
Effect of Eltanexor on the in vitro radiosensitivity of GBM cells in U87 (50 nM) and U251 (100 nM) cells. Viability was determined after either 14 (U87) (**A**) or 10 days (U251) (**C**). Left panel, fractional survival after irradiation in the presence of either solvent DMSO or Eltanexor (50 nM for U87 and 100 nM for U251 cells) at different doses of 0, 2, 4, and 6 Gy. The average dose enhancement factors (DEF) are provided for each cell line. Values are shown as mean ± S.D. from three independent experiments performed in triplicates. Statistical significance was determined by the Student’s *t*-test with *, *p* < 0.05 and ****, *p* < 0.0001. Right panel, colony formation assay demonstrating a significant effect of Eltanexor (50 nM and 100 nM) on radiation sensitivity in U87 and U251 cells. Scale bars in (**B**,**D**), 5 mm.

## Data Availability

Data presented in this study are available through the corresponding author upon written request.

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
