# Peer review of "Eltanexor Effectively Reduces Viability of Glioblastoma and Glioblastoma Stem-Like Cells at Nano-Molar Concentrations and Sensitizes to Radiotherapy and Temozolomide"

_biomedicines, 2022, doi:10.3390/biomedicines10092145_

Round 1

Reviewer 1 Report (Previous Reviewer 2)

The article entitled 'Eltanexor effectively reduces viability of glioblastoma and glioblastoma stem-like cells at nano-molar concentrations and sensitizes to radiotherapy and temozolomide’ describes the identification of a novel compound XPO1 inhibitor Eltanexor for the treatment of GBM. The article is very intriguing, well written, and provides compelling evidence after revision.

Reviewer 2 Report (Previous Reviewer 1)

The authors has addressed most concerns. Even though the in vivo data is still not provided, I do agree that at this stage, it should not prevent the paper being published. 

This manuscript is a resubmission of an earlier submission. The following is a list of the peer review reports and author responses from that submission.

Round 1

Reviewer 1 Report

In this manuscript, Otte et al., used Eltanexor which is a second-generation XPO1 nuclear export inhibitor on GBM cell lines and GSC cells. They showed that Eltanexor caused apoptosis at hundred nanomolar range. There may be also synergistic effect of combination therapy with irradiation.  In general, the design is straight forward, and used multiple cell lines and GSC models are a plus. However, there are a few major issues need to be addressed before acceptance.

1, Eltanexor is believed to be less brain-barrier permeable than Selinexor. This is a serious concern. In vivo models are needed.

2, Normal astrocytes and neural stem cells are needed as controls.

3, The claim that CDKN1A, P53 etc have nuclear accumulation is not supported. One does observe overall intensity increase in IF (same as in Western), but the ratio of nuclear vs. cytoplasm need to be calculated.

4. For the claim of synergy or additive, one needs to use Isobole graph.

5. What is the expression of XPO1 in these cancer cells versus normal controls?

Author Response

See Word Document in the attachment.

Reviewer 2 Report

The article entitled 'Eltanexor effectively reduces viability of glioblastoma and glioblastoma stem-like cells at nano-molar concentrations and sensitizes to radiotherapy and temozolomide’ describes the identification of a novel compound XPO1 inhibitor Eltanexor for the treatment of GBM. The article is very intriguing, well written, and provides compelling evidence.  However, The author could improve the article.

1.      The Figure 1a legends needs to describe which figures belong to the controls, as it is ambiguous.

2.      In Figure 2, The authors shows the effect of caspae activity in cell lines and patient derived material. The authors need to describe how they have quantified caspase activity in the description.

3.      Figure 2 needs to complemented with additional data such as immunoblots for caspase 3 or flow cytometry assays looking into apoptosis?

4.      It would be interesting to understand if the cell cycle is deregulated after treatment with the inhibitor, as the authors show the involvement of CDKN1A.

5.      Fig3 data is interesting, the authors investigated the expression of these proteins at the mRNA level, It would be interesting if the authors could quantify the immunoblots.

6.      The basal levels of p53 is very high, did the authors consider to serum starve the cells ?

7.      What the upstream receptors of p53 regulated by this pathway ?

8.      Can the authors probe for the regulators for this pathway as it is unclear how the biological roles such as invasion, migration are taking place ?

9.      Can the authors include a summary picture to give an overview of the findings.

10.  The need to include IHC stainings from patient derived materials.

11.  The discussion needs to include some of the articles related to the latest developments.

Author Response

See Word-Document in the attachment.
